# Filtering Variational Objectives

**Chris J. Maddison[1,3,\*], Dieterich Lawson,[2,\*] George Tucker[2,\*]**
**Nicolas Heess[1], Mohammad Norouzi[2], Andriy Mnih[1], Arnaud Doucet[3], Yee Whye Teh[1]**

[1]DeepMind, [2]Google Brain, [3]University of Oxford
{cmaddis, dieterichl, gjt}@google.com

## Abstract

When used as a surrogate objective for maximum likelihood estimation in latent variable models, the evidence lower bound (ELBO) produces state-of-the-art results. Inspired by this, we consider the extension of the ELBO to a family of lower bounds defined by a particle filter's estimator of the marginal likelihood, the *filtering variational objectives* (FIVOs). FIVOs take the same arguments as the ELBO, but can exploit a model's sequential structure to form tighter bounds. We present results that relate the tightness of FIVO's bound to the variance of the particle filter's estimator by considering the generic case of bounds defined as log-transformed likelihood estimators. Experimentally, we show that training with FIVO results in substantial improvements over training the same model architecture with the ELBO on sequential data.

## 1 Introduction

Learning in statistical models via gradient descent is straightforward when the objective function and its gradients are tractable. In the presence of latent variables, however, many objectives become intractable. For neural generative models with latent variables, there are currently a few dominant approaches: optimizing lower bounds on the marginal log-likelihood [1, 2], restricting to a class of invertible models [3], or using likelihood-free methods [4, 5, 6, 7]. In this work, we focus on the first approach and introduce *filtering variational objectives* (FIVOs), a tractable family of objectives for maximum likelihood estimation (MLE) in latent variable models with sequential structure.

Specifically, let $x$ denote an observation of an $\mathcal{X}$-valued random variable. We assume that the process generating $x$ involves an unobserved $\mathcal{Z}$-valued random variable $z$ with joint density $p(x, z)$ in some family $\mathcal{P}$. The goal of MLE is to recover $p \in \mathcal{P}$ that maximizes the marginal log-likelihood, $\log p(x) = \log \left( \int p(x, z) \, dz \right)$[1]. The difficulty in carrying out this optimization is that the log-likelihood function is defined via a generally intractable integral. To circumvent marginalization, a common approach [1, 2] is to optimize a variational lower bound on the marginal log-likelihood [8, 9]. The evidence lower bound $\mathcal{L}(x, p, q)$ (ELBO) is the most common such bound and is defined by a variational posterior distribution $q(z|x)$ whose support includes $p$'s,

$$\mathcal{L}(x, p, q) = \mathop{\mathbb{E}}_{q(z|x)} \left[ \log \frac{p(x, z)}{q(z|x)} \right] = \log p(x) - \mathrm{KL}(q(z|x) \parallel p(z|x)) \leq \log p(x) \,. \qquad (1)$$

$\mathcal{L}(x, p, q)$ lower-bounds the marginal log-likelihood for any choice of $q$, and the bound is tight when $q$ is the true posterior $p(z|x)$. Thus, the joint optimum of $\mathcal{L}(x, p, q)$ in $p$ and $q$ is the MLE. In practice, it is common to restrict $q$ to a tractable family of distributions (e.g., a factored distribution) and to

---

[\*]Equal contribution.
[1]We reuse $p$ to denote the conditionals and marginals of the joint density.

jointly optimize the ELBO over $p$ and $q$ with stochastic gradient ascent [1, 2, 10, 11]. Because of the KL penalty from $q$ to $p$, optimizing (1) under these assumptions tends to force $p$'s posterior to satisfy the factorizing assumptions of the variational family which reduces the capacity of the model $p$. One strategy for addressing this is to decouple the tightness of the bound from the quality of $q$. For example, [12] observed that Eq. (1) can be interpreted as the log of an unnormalized importance weight with the proposal given by $q$, and that using $N$ samples from the same proposal produces a tighter bound, known as the importance weighted auto-encoder bound, or IWAE.

Indeed, it follows from Jensen's inequality that the log of *any* unbiased positive Monte Carlo estimator of the marginal likelihood results in a lower bound that can be optimized for MLE. The filtering variational objectives (FIVOs) build on this idea by treating the log of a particle filter's likelihood estimator as an objective function. Following [13], we call objectives defined as log-transformed likelihood estimators Monte Carlo objectives (MCOs). In this work, we show that the tightness of an MCO scales like the relative variance of the estimator from which it is constructed. It is well-known that the variance of a particle filter's likelihood estimator scales more favourably than simple importance sampling for models with sequential structure [14, 15]. Thus, FIVO can potentially form a much tighter bound on the marginal log-likelihood than IWAE.

The main contributions of this work are introducing filtering variational objectives and a more careful study of Monte Carlo objectives. In Section 2, we review maximum likelihood estimation via maximizing the ELBO. In Section 3, we study Monte Carlo objectives and provide some of their basic properties. We define filtering variational objectives in Section 4, discuss details of their optimization, and present a sharpness result. Finally, we cover related work and present experiments showing that sequential models trained with FIVO outperform models trained with ELBO or IWAE in practice.

## 2  Background

We briefly review techniques for optimizing the ELBO as a surrogate MLE objective. We restrict our focus to latent variable models in which the model $p_\theta(x, z)$ factors into tractable conditionals $p_\theta(z)$ and $p_\theta(x|z)$ that are parameterized differentiably by parameters $\theta$. MLE in these models is then the problem of optimizing $\log p_\theta(x)$ in $\theta$. The expectation-maximization (EM) algorithm is an approach to this problem which can be seen as coordinate ascent, fully maximizing $\mathcal{L}(x, p_\theta, q)$ alternately in $q$ and $\theta$ at each iteration [16, 17, 18]. Yet, EM rarely applies in general, because maximizing over $q$ for a fixed $\theta$ corresponds to a generally intractable inference problem.

Instead, an approach with mild assumptions on the model is to perform gradient ascent following a Monte Carlo estimator of the ELBO's gradient [19, 10]. We assume that $q$ is taken from a family of distributions parameterized differentiably by parameters $\phi$. We can follow an unbiased estimator of the ELBO's gradient by sampling $z \sim q_\phi(z|x)$ and updating the parameters by $\theta' = \theta + \eta \nabla_\theta \log p_\theta(x, z)$ and $\phi' = \phi + \eta (\log p_\theta(x, z) - \log q_\phi(z|x)) \nabla_\phi \log q_\phi(z|x)$, where the gradients are computed conditional on the sample $z$ and $\eta$ is a learning rate. Such estimators follow the ELBO's gradient in expectation, but variance reduction techniques are usually necessary [10, 20, 13].

A lower variance gradient estimator can be derived if $q_\phi$ is a reparameterizable distribution [1, 2, 21]. Reparameterizable distributions are those that can be simulated by sampling from a distribution $\epsilon \sim d(\epsilon)$, which does not depend on $\phi$, and then applying a deterministic transformation $z = f_\phi(x, \epsilon)$. When $p_\theta$, $q_\phi$, and $f_\phi$ are differentiable, an unbiased estimator of the ELBO gradient consists of sampling $\epsilon$ and updating the parameter by $(\theta', \phi') = (\theta, \phi) + \eta \nabla_{(\theta, \phi)} (\log p_\theta(x, f_\phi(x, \epsilon)) - \log q_\phi(f_\phi(x, \epsilon)|x))$. Given $\epsilon$, the gradients of the sampling process can flow through $z = f_\phi(x, \epsilon)$.

Unfortunately, when the variational family of $q_\phi$ is restricted, following gradients of $-\mathrm{KL}(q_\phi(z|x) \,\|\, p_\theta(z|x))$ tends to reduce the capacity of the model $p_\theta$ to match the assumptions of the variational family. This KL penalty can be "removed" by considering generalizations of the ELBO whose tightness can be controlled by means other than the closenesss of $p$ and $q$, e.g., [12]. We consider this in the next section.

## 3  Monte Carlo Objectives (MCOs)

Monte Carlo objectives (MCOs) [13] generalize the ELBO to objectives defined by taking the $\log$ of a positive, unbiased estimator of the marginal likelihood. The key property of MCOs is that

they are lower bounds on the marginal log-likelihood, and thus can be used for MLE. Motivated by the previous section, we present results on the convergence of generic MCOs to the marginal log-likelihood and show that the tightness of an MCO is closely related to the variance of the estimator that defines it.

One can verify that the ELBO is a lower bound by using the concavity of log and Jensen's inequality,

$$\mathbb{E}_{q(z|x)} \left[ \log \frac{p(x,z)}{q(z|x)} \right] \leq \log \int \frac{p(x,z)}{q(z|x)} q(z|x) \, dz = \log p(x). \tag{2}$$

This argument only relies only on unbiasedness of $p(x,z)/q(z|x)$ when $z \sim q(z|x)$. Thus, we can generalize this by considering any unbiased marginal likelihood estimator $\hat{p}_N(x)$ and treating $\mathbb{E}[\log \hat{p}_N(x)]$ as an objective function over models $p$. Here $N \in \mathbb{N}$ indexes the amount of computation needed to simulate $\hat{p}_N(x)$, e.g., the number of samples or particles.

**Definition 1.** Monte Carlo Objectives. Let $\hat{p}_N(x)$ be an unbiased positive estimator of $p(x)$, $\mathbb{E}[\hat{p}_N(x)] = p(x)$, then the Monte Carlo objective $\mathcal{L}_N(x, p)$ over $p \in \mathcal{P}$ defined by $\hat{p}_N(x)$ is

$$\mathcal{L}_N(x, p) = \mathbb{E}[\log \hat{p}_N(x)] \tag{3}$$

For example, the ELBO is constructed from a single unnormalized importance weight $\hat{p}(x) = p(x,z)/q(z|x)$. The IWAE bound [12] takes $\hat{p}_N(x)$ to be $N$ averaged i.i.d. importance weights,

$$\mathcal{L}_N^{\text{IWAE}}(x, p, q) = \mathbb{E}_{q(z^i|x)} \left[ \log \left( \frac{1}{N} \sum_{i=1}^{N} \frac{p(x, z^i)}{q(z^i|x)} \right) \right] \tag{4}$$

We consider additional examples in the Appendix. To avoid notational clutter, we omit the arguments to an MCO, e.g., the observations $x$ or model $p$, when the default arguments are clear from context. Whether we can compute stochastic gradients of $\mathcal{L}_N$ efficiently depends on the specific form of the estimator and the underlying random variables that define it.

Many likelihood estimators $\hat{p}_N(x)$ converge to $p(x)$ almost surely as $N \to \infty$ (known as strong consistency). The advantage of a consistent estimator is that its MCO can be driven towards $\log p(x)$ by increasing $N$. We present sufficient conditions for this convergence and a description of the rate:

**Proposition 1.** Properties of Monte Carlo Objectives. *Let $\mathcal{L}_N(x, p)$ be a Monte Carlo objective defined by an unbiased positive estimator $\hat{p}_N(x)$ of $p(x)$. Then,*

   (a) *(Bound)* $\mathcal{L}_N(x, p) \leq \log p(x)$.

   (b) *(Consistency) If $\log \hat{p}_N(x)$ is uniformly integrable (see Appendix for definition) and $\hat{p}_N(x)$ is strongly consistent, then $\mathcal{L}_N(x, p) \to \log p(x)$ as $N \to \infty$.*

   (c) *(Asymptotic Bias) Let $g(N) = \mathbb{E}[(\hat{p}_N(x) - p(x))^6]$ be the 6th central moment. If the 1st inverse moment is bounded, $\limsup_{N \to \infty} \mathbb{E}[\hat{p}_N(x)^{-1}] < \infty$, then*

$$\log p(x) - \mathcal{L}_N(x, p) = \frac{1}{2} \text{var} \left( \frac{\hat{p}_N(x)}{p(x)} \right) + \mathcal{O}(\sqrt{g(N)}). \tag{5}$$

*Proof.* See the Appendix for the proof and a sufficient condition for controlling the first inverse moment when $\hat{p}_N(x)$ is the average of i.i.d. random variables. □

In some cases, convergence of the bound to $\log p(x)$ is monotonic, e.g., IWAE [12], but this is not true in general. The relative variance of estimators, $\text{var}(\hat{p}_N(x)/p(x))$, tends to be well studied, so property (c) gives us a tool for comparing the convergence rate of distinct MCOs. For example, [14, 15] study marginal likelihood estimators defined by particle filters and find that the relative variance of these estimators scales favorably in comparison to naive importance sampling. This suggests that a particle filter's MCO, introduced in the next section, will generally be a tighter bound than IWAE.

**Algorithm 1** Simulating $\mathcal{L}_N^{\text{FIVO}}(x_{1:T}, p, q)$

| | |
|---|---|
| 1: **FIVO**$(x_{1:T}, p, q, N)$: | 10:      **if** resampling criteria satisfied by $\{w_t^i\}_{i=1}^N$ **then** |
| 2: $\{w_0^i\}_{i=1}^N = \{1/N\}_{i=1}^N$ | 11:        $\{w_t^i, z_{1:t}^i\}_{i=1}^N = \textbf{RSAMP}(\{w_t^i, z_{1:t}^i\}_{i=1}^N)$ |
| 3: **for** $t \in \{1, \ldots, T\}$ **do** | 12: **return** $\log \hat{p}_N(x_{1:T})$ |
| 4:    **for** $i \in \{1, \ldots, N\}$ **do** | |
| 5:      $z_t^i \sim q_t(z_t \mid x_{1:t}, z_{1:t-1}^i)$ | 13: **RSAMP**$(\{w^i, z^i\}_{i=1}^N)$: |
| 6:      $z_{1:t}^i = \textbf{CONCAT}(z_{1:t-1}^i, z_t^i)$ | 14: **for** $i \in \{1, \ldots, N\}$ **do** |
| 7:    $\hat{p}_t = \left( \sum_{i=1}^N w_{t-1}^i \alpha_t(z_{1:t}^i) \right)$ | 15:    $a \sim \text{Categorical}(\{w^i\}_{i=1}^N)$ |
| 8:    $\hat{p}_N(x_{1:t}) = \hat{p}_N(x_{1:t-1}) \hat{p}_t$ | 16:    $y^i = z^a$ |
| 9:    $\{w_t^i\}_{i=1}^N = \{w_{t-1}^i \alpha_t(z_{1:t}^i)/\hat{p}_t\}_{i=1}^N$ | 17: **return** $\{\frac{1}{N}, y^i\}_{i=1}^N$ |

## 4   Filtering Variational Objectives (FIVOs)

The filtering variational objectives (FIVOs) are a family of MCOs defined by the marginal likelihood estimator of a particle filter. For models with sequential structure, e.g., latent variable models of audio and text, the relative variance of a naive importance sampling estimator tends to scale exponentially in the number of steps. In contrast, the relative variance of particle filter estimators can scale more favorably with the number of steps—linearly in some cases [14, 15]. Thus, the results of Section 3 suggest that FIVOs can serve as tighter objectives than IWAE for MLE in sequential models.

Let our observations be sequences of $T$ $\mathcal{X}$-valued random variables denoted $x_{1:T}$, where $x_{i:j} \equiv (x_i, \ldots, x_j)$. We also assume that the data generation process relies on a sequence of $T$ unobserved $\mathcal{Z}$-valued latent variables denoted $z_{1:T}$. We focus on sequential latent variable models that factor as a series of tractable conditionals, $p(x_{1:T}, z_{1:T}) = p_1(x_1, z_1) \prod_{t=2}^T p_t(x_t, z_t \mid x_{1:t-1}, z_{1:t-1})$.

A particle filter is a sequential Monte Carlo algorithm, which propagates a population of $N$ weighted particles for $T$ steps using a combination of importance sampling and resampling steps, see Alg. 1. In detail, the particle filter takes as arguments an observation $x_{1:T}$, the number of particles $N$, the model distribution $p$, and a variational posterior $q(z_{1:T} \mid x_{1:T})$ factored over $t$,

$$q(z_{1:T} \mid x_{1:T}) = \prod_{t=1}^T q_t(z_t \mid x_{1:t}, z_{1:t-1}) . \tag{6}$$

The particle filter maintains a population $\{w_{t-1}^i, z_{1:t-1}^i\}_{i=1}^N$ of particles $z_{1:t-1}^i$ with weights $w_{t-1}^i$. At step $t$, the filter independently proposes an extension $z_t^i \sim q_t(z_t \mid x_{1:t}, z_{1:t-1}^i)$ to each particle's trajectory $z_{1:t-1}^i$. The weights $w_{t-1}^i$ are multiplied by the incremental importance weights,

$$\alpha_t(z_{1:t}^i) = \frac{p_t(x_t, z_t^i \mid x_{1:t-1}, z_{1:t-1}^i)}{q_t(z_t^i \mid x_{1:t}, z_{1:t-1}^i)}, \tag{7}$$

and renormalized. If the current weights $w_t^i$ satisfy a resampling criteria, then a resampling step is performed and $N$ particles $z_{1:t}^i$ are sampled in proportion to their weights from the current population with replacement. Common resampling schemes include resampling at every step and resampling if the effective sample size (ESS) of the population $(\sum_{i=1}^N (w_t^i)^2)^{-1}$ drops below $N/2$ [22]. After resampling the weights are reset to 1. Otherwise, the particles $z_{1:t}^i$ are copied to the next step along with the accumulated weights. See Fig. 1 for a visualization.

Instead of viewing Alg. 1 as an inference algorithm, we treat the quantity $\mathbb{E}[\log \hat{p}_N(x_{1:T})]$ as an objective function over $p$. Because $\hat{p}_N(x_{1:T})$ is an unbiased estimator of $p(x_{1:T})$, proven in the Appendix and in [23, 24, 25, 26], it defines an MCO, which we call FIVO:

**Definition 2.** Filtering Variational Objectives. Let $\log \hat{p}_N(x_{1:T})$ be the output of Alg. 1 with inputs $(x_{1:T}, p, q, N)$, then $\mathcal{L}_N^{\text{FIVO}}(x_{1:T}, p, q) = \mathbb{E}[\log \hat{p}_N(x_{1:T})]$ is a filtering variational objective.

$\hat{p}_N(x_{1:T})$ is a strongly consistent estimator [23, 24]. So if $\log \hat{p}_N(x_{1:T})$ is uniformly integrable, then $\mathcal{L}_N^{\text{FIVO}}(x_{1:T}, p, q) \to \log p(x_{1:T})$ as $N \to \infty$. Resampling is the distinguishing feature of $\mathcal{L}_N^{\text{FIVO}}$; if resampling is removed, then FIVO reduces to IWAE. Resampling does add an amount of immediate variance, but it allows the filter to discard low weight particles with high probability. This has the

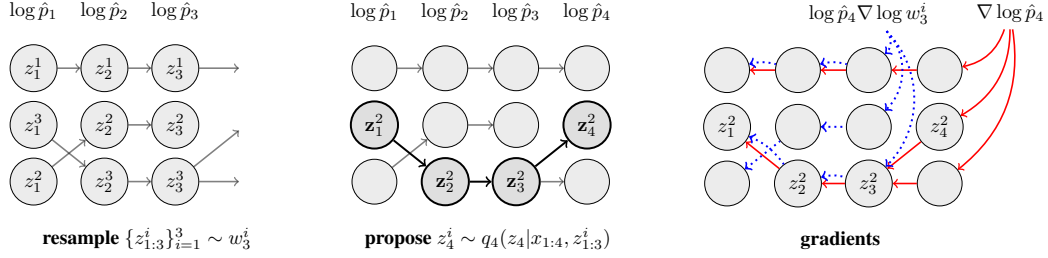

Figure 1: Visualizing FIVO; (Left) Resample from particle trajectories to determine inheritance in next step, (middle) propose with $q_t$ and accumulate loss $\log \hat{p}_t$, (right) gradients (in the reparameterized case) flow through the lattice, objective gradients in solid red and resampling gradients in dotted blue.

effect of refocusing the distribution of particles to regions of higher mass under the posterior, and in some sequential models can reduce the variance from exponential to linear in the number of time steps [14, 15]. Resampling is a greedy process, and it is possible that a particle discarded at step $t$, could have attained a high mass at step $T$. In practice, the best trade-off is to use adaptive resampling schemes [22]. If for a given $x_{1:T}, p, q$ a particle filter's likelihood estimator improves over simple importance sampling in terms of variance, we expect $\mathcal{L}_N^{\text{FIVO}}$ to be a tighter bound than $\mathcal{L}$ or $\mathcal{L}_N^{\text{IWAE}}$.

## 4.1 Optimization

The FIVO bound can be optimized with the same stochastic gradient ascent framework used for the ELBO. We found in practice it was effective simply to follow a Monte Carlo estimator of the biased gradient $\mathbb{E}[\nabla_{(\theta,\phi)} \log \hat{p}_N(x_{1:T})]$ with reparameterized $z_t^i$. This gradient estimator is biased, as the full FIVO gradient has three kinds of terms: it has the term $\mathbb{E}[\nabla_{\theta,\phi} \log \hat{p}_N(x_{1:T})]$, where $\nabla_{\theta,\phi} \log \hat{p}_N(x_{1:T})$ is defined conditional on the random variables of Alg. 1; it has gradient terms for every distribution of Alg. 1 that depends on the parameters; and, if adaptive resampling is used, then it has additional terms that account for the change in FIVO with respect to the decision to resample. In this section, we derive the FIVO gradient when $z_t^i$ are reparameterized and a fixed resampling schedule is followed. We derive the full gradient in the Appendix.

In more detail, we assume that $p$ and $q$ are parameterized in a differentiable way by $\theta$ and $\phi$. Assume that $q$ is from a reparameterizable family and that $z_t^i$ of Alg. 1 are reparameterized. Assume that we use a fixed resampling schedule, and let $\mathbb{I}(\text{resampling at step } t)$ be an indicator function indicating whether a resampling occured at step $t$. Now, $\mathcal{L}_N^{\text{FIVO}}$ depends on the parameters via $\log \hat{p}_N(x_{1:T})$ and the resampling probabilities $w_t^i$ in the density. Thus, $\nabla_{(\theta,\phi)} \mathcal{L}_N^{\text{FIVO}} =$

$$\mathbb{E}\left[\nabla_{(\theta,\phi)} \log \hat{p}_N(x_{1:T}) + \sum_{t=1}^{T} \sum_{i=1}^{N} \mathbb{I}(\text{resampling at step } t) \log \frac{\hat{p}_N(x_{1:T})}{\hat{p}_N(x_{1:t})} \nabla_{(\theta,\phi)} \log w_t^i\right] \quad (8)$$

Given a single forward pass of Alg. 1 with reparameterized $z_t^i$, the terms inside the expectation form a Monte Carlo estimator of Eq. (8). However, the terms from resampling events contribute to the majority of the variance of the estimator. Thus, the gradient estimator that we found most effective in practice consists only of the gradient $\nabla_{(\theta,\phi)} \log \hat{p}_N(x_{1:T})$, the solid red arrows of Figure 1. We explore this experimentally in Section 6.3.

## 4.2 Sharpness

As with the ELBO, FIVO is a variational objective taking a variational posterior $q$ as an argument. An important question is whether FIVO achieves the marginal log-likelihood at its optimal $q$. We can only guarantee this for models in which $z_{1:t-1}$ and $x_t$ are independent given $x_{1:t-1}$.

**Proposition 2.** Sharpness of Filtering Variational Objectives. *Let $\mathcal{L}_N^{\text{FIVO}}(x_{1:T}, p, q)$ be a FIVO, and $q^*(x_{1:T}, p) = \arg\max_q \mathcal{L}_N^{\text{FIVO}}(x_{1:T}, p, q)$. If $p$ has independence structure such that $p(z_{1:t-1}|x_{1:t}) = p(z_{1:t-1}|x_{1:t-1})$ for $t \in \{2, \ldots, T\}$, then*

$$q^*(x_{1:T}, p)(z_{1:T}) = p(z_{1:T}|x_{1:T}) \quad \text{and} \quad \mathcal{L}_N^{\text{FIVO}}(x_{1:T}, p, q^*(x_{1:T}, p)) = \log p(x_{1:T}).$$

*Proof.* See Appendix. □

Most models do not satisfy this assumption, and deriving the optimal $q$ in general is complicated by the resampling dynamics. For the restricted the model class in Proposition 2, the optimal $q_t$ does not condition on future observations $x_{t+1:T}$. We explored this experimentally with richer models in Section 6.4, and found that allowing $q_t$ to condition on $x_{t+1:T}$ does not reliably improve FIVO. This is consistent with the view of resampling as a greedy process that responds to each intermediate distribution as if it were the final. Still, we found that the impact of this effect was outweighed by the advantage of optimizing a tighter bound.

## 5  Related Work

The marginal log-likelihood is a central quantity in statistics and probability, and there has long been an interest in bounding it [27]. The literature relating to the bounds we call Monte Carlo objectives has typically focused on the problem of estimating the marginal likelihood itself. [28, 29] use Jensen's inequality in a forward and reverse estimator to detect the failure of inference methods. IWAE [12] is a clear influence on this work, and FIVO can be seen as an extension of this bound. The ELBO enjoys a long history [8] and there have been efforts to improve the ELBO itself. [30] generalize the ELBO by considering arbitrary operators of the model and variational posterior. More closely related to this work is a body of work improving the ELBO by increasing the expressiveness of the variational posterior. For example, [31, 32] augment the variational posterior with deterministic transformations with fixed Jacobians, and [33] extend the variational posterior to admit a Markov chain.

Other approaches to learning in neural latent variable models include [34], who use importance sampling to approximate gradients under the posterior, and [35], who use sequential Monte Carlo to approximate gradients under the posterior. These are distinct from our contribution in the sense that for them inference for the sake of estimation is the ultimate goal. To our knowledge the idea of treating the output of inference as an objective in and of itself, while not completely novel, has not been fully appreciated in the literature. Although, this idea shares inspiration with methods that optimize the convergence of Markov chains [36].

We note that the idea to optimize the log estimator of a particle filter was independently and concurrently considered in [37, 38]. In [37] the bound we call FIVO is cast as a tractable lower bound on the ELBO defined by the particle filter's non-parameteric approximation to the posterior. [38] additionally derive an expression for FIVO's bias as the KL between the filter's distribution and a certain target process. Our work is distinguished by our study of the convergence of MCOs in $N$, which includes FIVO, our investigation of FIVO sharpness, and our experimental results on stochastic RNNs.

## 6  Experiments

In our experiments, we sought to: (a) compare models trained with ELBO, IWAE, and FIVO bounds in terms of final test log-likelihoods, (b) explore the effect of the resampling gradient terms on FIVO, (c) investigate how the lack of sharpness affects FIVO, and (d) consider how models trained with FIVO use the stochastic state. To explore these questions, we trained variational recurrent neural networks (VRNN) [39] with the ELBO, IWAE, and FIVO bounds using TensorFlow [40] on two benchmark sequential modeling tasks: natural speech waveforms and polyphonic music. These datasets are known to be difficult to model without stochastic latent states [41].

The VRNN is a sequential latent variable model that combines a deterministic recurrent neural network (RNN) with stochastic latent states $z_t$ at each step. The observation distribution over $x_t$ is conditioned directly on $z_t$ and indirectly on $z_{1:t-1}$ via the RNN's state $h_t(z_{t-1}, x_{t-1}, h_{t-1})$. For a length $T$ sequence, the model's posterior factors into the conditionals $\prod_{t=1}^{T} p_t(z_t|h_t(z_{t-1}, x_{t-1}, h_{t-1}))g_t(x_t|z_t, h_t(z_{t-1}, x_{t-1}, h_{t-1}))$, and the variational posterior factors as $\prod_{t=1}^{T} q_t(z_t|h_t(z_{t-1}, x_{t-1}, h_{t-1}), x_t)$. All distributions over latent variables are factorized Gaussians, and the output distributions $g_t$ depend on the dataset. The RNN is a single-layer LSTM and the conditionals are parameterized by fully connected neural networks with one hidden layer of the same size as the LSTM hidden layer. We used the residual parameterization [41] for the variational posterior.

| $N$ | Bound | Nottingham | JSB | MuseData | Piano-midi.de | $N$ | Bound | TIMIT 64 units | TIMIT 256 units |
|---|---|---|---|---|---|---|---|---|---|
| 4 | ELBO | -3.00 | -8.60 | -7.15 | -7.81 | 4 | ELBO | 0 | 10,438 |
| | IWAE | -2.75 | -7.86 | -7.20 | -7.86 | | IWAE | -160 | 11,054 |
| | FIVO | **-2.68** | **-6.90** | **-6.20** | **-7.76** | | FIVO | **5,691** | **17,822** |
| 8 | ELBO | -3.01 | -8.61 | -7.19 | -7.83 | 8 | ELBO | 2,771 | 9,819 |
| | IWAE | -2.90 | -7.40 | -7.15 | -7.84 | | IWAE | 3,977 | 11,623 |
| | FIVO | **-2.77** | **-6.79** | **-6.12** | **-7.45** | | FIVO | **6,023** | **21,449** |
| 16 | ELBO | -3.02 | -8.63 | -7.18 | -7.85 | 16 | ELBO | 1,676 | 9,918 |
| | IWAE | -2.85 | -7.41 | -7.13 | -7.79 | | IWAE | 3,236 | 13,069 |
| | FIVO | **-2.58** | **-6.72** | **-5.89** | **-7.43** | | FIVO | **8,630** | **21,536** |

Table 1: Test set marginal log-likelihood bounds for models trained with ELBO, IWAE, and FIVO. For ELBO and IWAE models, we report $\max\{\mathcal{L}, \mathcal{L}_{128}^{\mathrm{IWAE}}, \mathcal{L}_{128}^{\mathrm{FIVO}}\}$. For FIVO models, we report $\mathcal{L}_{128}^{\mathrm{FIVO}}$. Pianoroll results are in nats per timestep, TIMIT results are in nats per sequence relative to ELBO with $N = 4$. For details on our evaluation methodology and absolute numbers see the Appendix.

For FIVO we resampled when the ESS of the particles dropped below $N/2$. For FIVO and IWAE we used a batch size of 4, and for the ELBO, we used batch sizes of $4N$ to match computational budgets (resampling is $\mathcal{O}(N)$ with the alias method). For all models we report bounds using the variational posterior trained jointly with the model. For models trained with FIVO we report $\mathcal{L}_{128}^{\mathrm{FIVO}}$. To provide strong baselines, we report the maximum across bounds, $\max\{\mathcal{L}, \mathcal{L}_{128}^{\mathrm{IWAE}}, \mathcal{L}_{128}^{\mathrm{FIVO}}\}$, for models trained with ELBO and IWAE. Additional details in the Appendix.

## 6.1 Polyphonic Music

We evaluated VRNNs trained with the ELBO, IWAE, and FIVO bounds on 4 polyphonic music datasets: the Nottingham folk tunes, the JSB chorales, the MuseData library of classical piano and orchestral music, and the Piano-midi.de MIDI archive [42]. Each dataset is split into standard train, valid, and test sets and is represented as a sequence of 88-dimensional binary vectors denoting the notes active at the current timestep. We mean-centered the input data and modeled the output as a set of 88 factorized Bernoulli variables. We used 64 units for the RNN hidden state and latent state size for all polyphonic music models except for JSB chorales models, which used 32 units. We report bounds on average log-likelihood per timestep in Table 1. Models trained with the FIVO bound significantly outperformed models trained with either the ELBO or the IWAE bounds on all four datasets. In some cases, the improvements exceeded 1 nat *per timestep*, and in all cases optimizing FIVO with $N = 4$ outperformed optimizing IWAE or ELBO for $N = \{4, 8, 16\}$.

## 6.2 Speech

The TIMIT dataset is a standard benchmark for sequential models that contains 6300 utterances with an average duration of 3.1 seconds spoken by 630 different speakers. The 6300 utterances are divided into a training set of size 4620 and a test set of size 1680. We further divided the training set into a validation set of size 231 and a training set of size 4389, with the splits exactly as in [41]. Each TIMIT utterance is represented as a sequence of real-valued amplitudes which we split into a sequence of 200-dimensional frames, as in [39, 41]. Data preprocessing was limited to mean centering and variance normalization as in [41]. For TIMIT, the output distribution was a factorized Gaussian, and we report the average log-likelihood bound per sequence relative to models trained with ELBO. Again, models trained with FIVO significantly outperformed models trained with IWAE or ELBO, see Table 1.

## 6.3 Resampling Gradients

All models in this work (except those in this section) were trained with gradients that did not include the term in Eq. (8) that comes from resampling steps. We omitted this term because it has an outsized effect on gradient variance, often increasing it by 6 orders of magnitude. To explore the effects of this term experimentally, we trained VRNNs with and without the resampling gradient term on the TIMIT and polyphonic music datasets. When using the resampling term, we attempted to control its variance

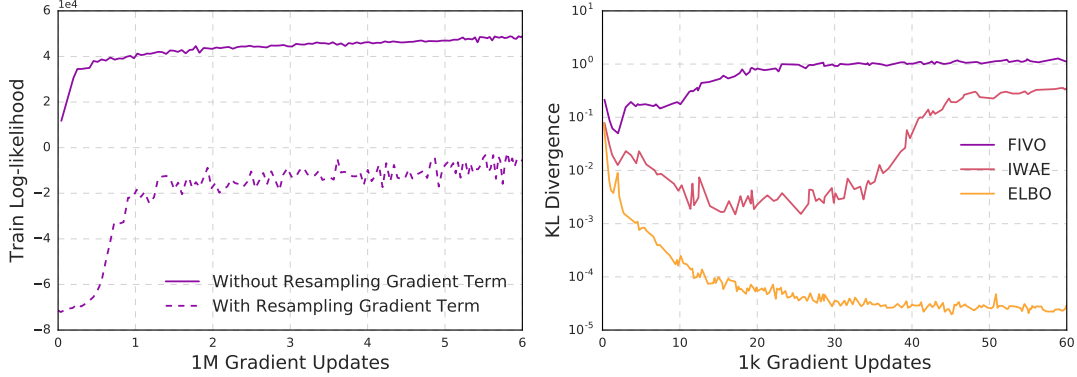

Figure 2: (Left) Graph of $\mathcal{L}_{128}^{\text{FIVO}}$ over training comparing models trained with and without the resampling gradient terms on TIMIT with $N = 4$. (Right) KL divergence from $q(z_{1:T}|x_{1:T})$ to $p(z_{1:T})$ for models trained on the JSB chorales with $N = 16$.

| Bound | Nottingham | JSB | MuseData | Piano-midi.de | TIMIT |
|---|---|---|---|---|---|
| ELBO | **-2.40** | **-5.48** | -6.54 | **-6.68** | **0** |
| ELBO+s | -2.59 | -5.53 | **-6.48** | -6.77 | -925 |
| IWAE | -2.52 | -5.77 | -6.54 | **-6.74** | 1,469 |
| IWAE+s | **-2.37** | **-4.63** | **-6.47** | **-6.74** | 2,630 |
| FIVO | **-2.29** | -4.08 | **-5.80** | -6.41 | 6,991 |
| FIVO+s | -2.34 | **-3.83** | -5.87 | **-6.34** | 9,773 |

Table 2: Train set marginal log-likelihood bounds for models comparing smoothing (+s) and non-smoothing variational posteriors. We report $\max\{\mathcal{L}, \mathcal{L}_{128}^{\text{IWAE}}, \mathcal{L}_{128}^{\text{FIVO}}\}$ for ELBO and IWAE models and $\mathcal{L}_{128}^{\text{FIVO}}$ for FIVO models. All models were trained with $N = 4$. Pianoroll results are in nats per timestep, TIMIT results are in nats per sequence relative to non-smoothing ELBO. For details on our evaluation methodology and absolute numbers see the Appendix.

using a moving-average baseline linear in the number of timesteps. For all datasets, models trained without the resampling gradient term outperformed models trained with the term by a large margin on both the training set and held-out data. Many runs with resampling gradients failed to improve beyond random initialization. A representative pair of train log-likelihood curves is shown in Figure 2 — gradients without the resampling term led to earlier convergence and a better solution. We stress that this is an empirical result — in principle biased gradients can lead to divergent behaviour. We leave exploring strategies to reduce the variance of the unbiased estimator to future work.

### 6.4 Sharpness

FIVO does not achieve the marginal log-likelihood at its optimal variational posterior $q^*$, because the optimal $q^*$ does not condition on future observations (see Section 4.2). In contrast, ELBO and IWAE are sharp, and their $q^*$s depend on future observations. To investigate the effects of this, we defined a smoothing variant of the VRNN in which $q$ takes as additional input the hidden state of a deterministic RNN run backwards over the observations, allowing $q$ to condition on future observations. We trained smoothing VRNNs using ELBO, IWAE, and FIVO, and report evaluation on the training set (to isolate the effect on optimization performance) in Table 2 . Smoothing helped models trained with IWAE, but not enough to outperform models trained with FIVO. As expected, smoothing did not reliably improve models trained with FIVO. Test set performance was similar, see the Appendix for details.

### 6.5 Use of Stochastic State

A known pathology when training stochastic latent variable models with the ELBO is that stochastic states can go unused. Empirically, this is associated with the collapse of variational posterior $q(z|x)$ network to the model prior $p(z)$ [43]. To investigate this, we plot the KL divergence from $q(z_{1:T}|x_{1:T})$ to $p(z_{1:T})$ averaged over the dataset (Figure 2). Indeed, the KL of models trained with

ELBO collapsed during training, whereas the KL of models trained with FIVO remained high, even while achieving a higher log-likelihood bound.

## 7 Conclusions

We introduced the family of filtering variational objectives, a class of lower bounds on the log marginal likelihood that extend the evidence lower bound. FIVOs are suited for MLE in neural latent variable models. We trained models with the ELBO, IWAE, and FIVO bounds and found that the models trained with FIVO significantly outperformed other models across four polyphonic music modeling tasks and a speech waveform modeling task. Future work will include exploring control variates for the resampling gradients, FIVOs defined by more sophisticated filtering algorithms, and new MCOs based on differentiable operators like leapfrog operators with deterministically annealed temperatures. In general, we hope that this paper inspires the machine learning community to take a fresh look at the literature of marginal likelihood estimators—seeing them as objectives instead of algorithms for inference.

**Acknowledgments**

We thank Matt Hoffman, Matt Johnson, Danilo J. Rezende, Jascha Sohl-Dickstein, and Theophane Weber for helpful discussions and support in this project. A. Doucet was partially supported by the EPSRC grant EP/K000276/1. Y. W. Teh's research leading to these results has received funding from the European Research Council under the European Union's Seventh Framework Programme (FP7/2007-2013) ERC grant agreement no. 617071.

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
