[Supplementary Material]

## Appendix to Filtering Variational Objectives

**Other Examples of MCOs.**

There is an extensive literature on marginal likelihood estimators [44, 45, 46, 47, 48, 49, 50]. Each defines an MCO, and we consider two in more detail, annealed importance sampling [48] and multiple importance sampling [44, 51]. Let $x$ denote an observation of an $\mathcal{X}$-valued random variable generated in a process with an unobserved $\mathcal{Z}$-valued random variable $z$. Let $p(x, z)$ be the joint density.

**Annealed Importance Sampling MCO.** Annealed importance sampling (AIS) is a generalization of importance sampling [48]. We present an MCO derived from a special case of the AIS algorithm. Let $q(z|x)$ be a variational posterior distribution and let $\beta_i$ be a sequence of real numbers for $i \in \{1, \dots, N+1\}$ such that $0 \le \beta_i \le 1$ and $\beta_1 = 0$ and $\beta_{N+1} = 1$. Let $T_i(z'|z, x)$ be a Markov transition distribution whose stationary distribution is proportional to $q(z|x)^{1-\beta_i} p(x, z)^{\beta_i}$. Then for $z_1 \sim q(z|x)$ and $z_i \sim T_i(z'|z_{i-1}, x)$ for $i \in \{2, \dots, N\}$ we have the following unbiased estimator,

$$\mathbb{E}[\hat{p}_N(x)] = \mathbb{E}\left[\prod_{i=1}^{N}\left(\frac{p(x, z_i)}{q(z_i|x)}\right)^{\beta_{i+1}-\beta_i}\right] = p(x) \tag{9}$$

Notice two things. First, there is no assumption that the states $z_i$ are at equilibrium, and second, we did not require a transition operator keeping $p(x, z)$ as an invariant distribution. All together, we can define the AIS MCO,

$$\mathcal{L}_{\mathrm{N}}^{\mathrm{AIS}}(x, q, \{T_i\}_{i=2}^{N}, p) = \mathbb{E}\left[\sum_{i=1}^{N}(\beta_{i+1} - \beta_i)\log\frac{p(x, z_i)}{q(z_i|x)}\right] \tag{10}$$

This is a sharp objective, if we take $q$ as the true posterior, $q(z|x) = p(z|x)$, and $T_i(z'|z, x) = \delta(z' - z)$ to be the Dirac delta copy operator. The difficulty in applying this MCO is finding $T_i$, which are scalable and easy to optimize. Generalizations of the AIS procedure have been proposed in [52]. The resulting Sequential Monte Carlo samplers procedures also provide an unbiased estimator of the marginal likelihood and are structurally identical to the particle algorithm presented in this paper.

**Multiple Importance Sampling MCO.** Multiple importance sampling (MIS) [44] is another generalization of importance sampling. Let $q_i(z|x)$ be $N$ possibly distinct variational posterior distributions and $w_i(x) \ge 0$ be such that $\sum_{i=1}^{N} w_i(x) = 1$. There are a variety of distinct estimators that could be formed from the $q_i$ [51]. We present just one. Let $z_i \sim q_i(z|x)$, then we have the following unbiased estimator

$$\mathbb{E}[\hat{p}_N(x)] = \mathbb{E}\left[\sum_{i=1}^{N}\frac{w_i(x)p(x, z_i)}{\sum_{j=1}^{N} w_j(x)q_j(z_i|x)}\right] = p(x) \tag{11}$$

Notice that the latent sample $z_i \sim q_i(z|x)$ is evaluated under all $q_i$'s. One can view this as a Rao-Blackwellized estimator corresponding to the mixture distribution $\sum_{i=1}^{N} w_i(x)q_i(z|x)$. All together,

$$\mathcal{L}_{\mathrm{N}}^{\mathrm{MIS}}(x, \{q_i\}_{i=1}^{N}, \{w_i\}_{i=1}^{N}, p) = \mathbb{E}\left[\log\left(\sum_{i=1}^{N}\frac{w_i(x)p(x, z_i)}{\sum_{j=1}^{N} w_j(x)q_j(z_i|x)}\right)\right] \tag{12}$$

Again, this objective is sharp, if we take any $q_i(z|x) = p(z|x)$ and $w_i(x) = 1$. The difficulty in making this objective more useful is optimizing it in a way that distinguishes the $q_i$ and assigns the appropriate $w_i$.

**Proof of Proposition 1.**

Let $\mathbb{E}[\hat{p}_N(x)] = p(x)$ and define $\mathcal{L}_N(x, p) = \mathbb{E}[\log \hat{p}_N(x)]$ as the Monte Carlo objective defined by $\hat{p}_N(x)$.

    (a) By the concavity of $\log$ and Jensen's inequality,

$$\mathcal{L}_N(x, p) = \mathbb{E}[\log \hat{p}_N(x)] \le \log \mathbb{E}[\hat{p}_N(x)] = \log p(x)$$

(b) Assume

- $\hat{p}_N(x)$ is strongly consistent, i.e. $\hat{p}_N(x) \xrightarrow{a.s.} p(x)$ as $N \to \infty$.
- $\log \hat{p}_N(x)$ is uniformly integrable. That is, let $(\Omega, \mathcal{F}, \mu)$ be the probability space on which $\log \hat{p}_N(x)$ is defined. The random variables $\{\log \hat{p}_N(x)\}_{N=1}^{\infty}$ are uniformly integrable if $\mathbb{E}[|\log \hat{p}_N(x)|] < \infty$ and if for any $\epsilon > 0$, there exists $\delta > 0$, such that for all $N$ and $E \in \mathcal{F}$, $\mu(E) < \delta$ implies $\mathbb{E}[|\log \hat{p}_N(x)|\mathbb{I}(E)] < \epsilon$, where $\mathbb{I}(E)$ is an indicator function of the set $E$.

Then by continuity of $\log$, $\log \hat{p}_N(x)$ converges almost surely to $\log p(x)$. By Vitali's convergence theorem (using the uniform integrability assumption), we get $\mathcal{L}_N(x, p) = \mathbb{E}[\log \hat{p}_N(x)] \to \log p(x)$ as $N \to \infty$.

(c) Let $g(N) = \mathbb{E}[(\hat{p}_N(x) - p(x))^6]$, and assume $\limsup_{N \to \infty} \mathbb{E}[(\hat{p}_N(x))^{-1}] < \infty$. Define the relative error

$$\Delta = \frac{\hat{p}_N(x) - p(x)}{p(x)} \tag{13}$$

Then the bias $\log p(x) - \mathcal{L}_N(x, p) = -\mathbb{E}[\log(1 + \Delta)]$. Now, Taylor expand $\log(1 + \Delta)$ about 0,

$$\log(1 + \Delta) = \Delta - \frac{1}{2}\Delta^2 + \int_0^{\Delta} \left( \frac{1}{1 + x} - 1 + x \right) dx \tag{14}$$

$$= \Delta - \frac{1}{2}\Delta^2 + \int_0^{\Delta} \left( \frac{x^2}{1 + x} \right) dx \tag{15}$$

and in expectation

$$-\mathbb{E}[\log(1 + \Delta)] = \frac{1}{2}\Delta^2 - \mathbb{E}\left[ \int_0^{\Delta} \left( \frac{x^2}{1 + x} \right) dx \right] \tag{16}$$

Our aim is to show

$$\left| \mathbb{E}\left[ \int_0^{\Delta} \frac{x^2}{1 + x} dx \right] \right| \in \mathcal{O}(g(N)^{1/2}) \tag{17}$$

In particular, by Cauchy-Schwarz

$$\left| \mathbb{E}\left[ \int_0^{\Delta} \left( \frac{x^2}{1 + x} \right) dx \right] \right| \le \mathbb{E}\left[ \left| \int_0^{\Delta} \frac{1}{(1 + x)^2} dx \right|^{1/2} \left| \int_0^{\Delta} x^4 dx \right|^{1/2} \right] \tag{18}$$

$$= \mathbb{E}\left[ \left| \frac{\Delta}{1 + \Delta} \right|^{1/2} \left| \frac{\Delta^5}{5} \right|^{1/2} \right] \tag{19}$$

$$= \mathbb{E}\left[ \left| \frac{1}{1 + \Delta} \right|^{1/2} \left| \frac{\Delta^6}{5} \right|^{1/2} \right] \tag{20}$$

and again by Cauchy-Schwarz

$$\le \left( \mathbb{E}\left[ \left| \frac{1}{1 + \Delta} \right| \right] \right)^{1/2} \left( \mathbb{E}\left[ \frac{\Delta^6}{5} \right] \right)^{1/2}. \tag{21}$$

This concludes the proof.

**Controlling the first inverse moment.**

We provide a sufficient condition that guarantees that the inverse moment of the average of i.i.d. random variables is bounded, a condition used in Proposition 1 (c). Intuitively, this is a fairly weak condition, because it only requires that the mass in an arbitrarily small neighbourhood of zero is bounded.

**Lemma 3.** *Let $w_i$ be i.i.d. positive random variables and $\hat{p}_N(x) = \frac{1}{N}\sum_{i=1}^{N} w_i$. If there exist $M, C, \epsilon > 0$ such that $\mathbb{P}(w_i < w) \leq Cw^{1+\epsilon}$ for $w \in [0, M)$, then $\mathbb{E}[\hat{p}_N(x)^{-1}] \leq C\frac{M^\epsilon}{\epsilon} + \frac{1}{M}$.*

*Proof.* Let $M, C, \epsilon > 0$ be such that $\mathbb{P}(w_i < w) \leq Cw^{1+\epsilon}$ for $w \in [0, M)$. We proceed in two cases. If $N = 1$, then

$$\mathbb{E}[\hat{p}_N(x)^{-1}] = \int_0^\infty \mathbb{P}(w_1^{-1} > u)\, du$$

$$= \int_0^\infty \mathbb{P}(w_1 < 1/u)\, du$$

$$= \int_0^M \frac{\mathbb{P}(w_1 < w)}{w^2}\, dw + \int_M^\infty \frac{\mathbb{P}(w_1 < w)}{w^2}\, dw$$

$$\leq \int_0^M \frac{Cw^{1+\epsilon}}{w^2}\, dw + \int_M^\infty \frac{1}{w^2}\, dw$$

$$= C\frac{M^\epsilon}{\epsilon} + \frac{1}{M}$$

For $N > 1$, we show that $\mathbb{E}[\hat{p}_N(x)^{-1}] \leq \mathbb{E}[\hat{p}_1(x)^{-1}]$, so the same condition is sufficient for any $N$. The AM-GM inequality tells us that

$$\sum_{i=1}^N \frac{w_i}{N} \geq \left(\prod_{i=1}^N w_i\right)^{1/N}$$

so

$$\mathbb{E}[\hat{p}_N(x)^{-1}] \leq \mathbb{E}\left[\left(\prod_{i=1}^N w_i\right)^{-1/N}\right]$$

$$= \prod_{i=1}^N \mathbb{E}\left[w_i^{-1/N}\right]$$

$$= \mathbb{E}\left[w_1^{-1/N}\right]^N$$

and by Lyapunov's inequality, we have

$$\leq \mathbb{E}\left[\left(w_1^{-1/N}\right)^N\right] = \mathbb{E}[\hat{p}_1(x)^{-1}]$$

This concludes the proof. □

**Unbiasedness of $\hat{p}_N(x_{1:T})$ from the particle filter.**

We sketch an argument that the random variable $\hat{p}_N(x_{1:T})$ defined by Algorithm 1 is an unbiased estimator of the marginal likelihood $p(x_{1:T})$. This is a well-known fact [23, 52, 25, 26], and our sketch is based on [25]. The strategy is to cast the particle filter's estimator $\hat{p}_N(x_{1:T})$ as a single importance weight over an extended space. The lack of bias in the particle filter therefore reduces to the unbiasedness of importance sampling. Key to this is identifying the target and proposal distributions in the extended space. The target distribution is called "conditional sequential Monte Carlo", Algorithm 2. The proposal distribution is the particle filter itself, Algorithm 1.

We argue that it is enough to consider just an arbitrary fixed (non-adaptive) resampling schedule that always resamples at step $T$. First, consider adaptive resampling criteria, i.e. criteria that are deterministic functions of the weights $w_t^i$. For such criteria the joint density of random variables in Algorithm 1 will be piecewise continuous, composed of $2^T$ regions corresponding to a sequence of resample/no-resample decisions. This density has a form on each piece that is exactly the same as the density for some fixed resampling schedule. Moreover, it is globally normalized, because of the sequential structure of the filter. Because Algorithm 2 makes the same decisions, it also is partitioned along the same sets and each piece has the same fixed resampling schedule. Thus, it is

---

**Algorithm 2** Conditional SMC

---

1: **CSMC**$(x_{1:T}, p, q, N)$:
2: $y_{1:T} \sim p(z_{1:T}|x_{1:T})$
3: $j = 1$
4: $\{w_0^i\}_{i=1}^N = \{1/N\}_{i=1}^N$
5: **for** $t \in \{1, \ldots, T\}$ **do**
6:     $z_{1:t}^j = y_{1:t}$
7:     **for** $i \neq j$ **do**
8:         $z_t^i \sim q_t(z_t|x_{1:t}, z_{1:t-1}^i)$
9:         $z_{1:t}^i = \textbf{CONCAT}(z_{1:t-1}^i, z_t^i)$

10:     $\hat{p}_t = \left(\sum_{i=1}^N w_{t-1}^i \alpha_t(z_{1:t}^i)\right)$
11:     $\{w_t^i\}_{i=1}^N = \{w_{t-1}^i \alpha_t(z_{1:t}^i)/\hat{p}_t\}_{i=1}^N$
12:     **if** resampling criteria satisfied by $\{w_t^i\}_{i=1}^N$ **then**
13:         $\{w_t^i, z_{1:t}^i\}_{i=1}^N = \textbf{RSAMP}(\{w_t^i, z_{1:t}^i\}_{i=1}^N)$
14:         $j \sim \text{Uniform}\{1, \ldots, N\}$
15:         $z_{1:t}^j = y_{1:t}$

---

enough to consider only a fixed resampling schedule. Second, notice that in the final step, step $T$, of Algorithms 1 and 2 resampling has no effect on $\hat{p}_N(x_{1:T})$. Thus, we assume that the resampling criteria of Algorithms 1 and 2 at step $T$ is set to always resample. All together it is safe to assume a fixed resampling schedule with $R$ resampling events, $1 \leq R \leq T$, at steps $k_r \in \{1, \ldots, T\}$ for $r \in \{0, \ldots, R\}$ with $k_R = T$ and $k_0 = 0$.

Now we derive the joint density of Algorithm 1 and 2 taken at each iteration *after* possibly resampling. To avoid notational clutter we let $g, f$ (omitting their arguments) represent the densities of the variables in Algorithms 1 and 2. Technically, we should also be keeping track of the indices that indicate the inheritance of the resampling step. So, let the random variables $\{\{w_t^i, z_{1:t}^i\}_{i=1}^N\}_{t=1}^T$ be the particles *before* resampling and $s(i) \in \{1, \ldots, N\}$ be the index that is selected for inheritance of the $i$th particle *after* resampling. Then the density corresponding to Algorithm 1 is

$$g = \prod_{r=1}^R \prod_{i=1}^N w_{k_r}^{s(i)} \prod_{k=k_{r-1}+1}^{k_r} q_k(z_k^i|x_{1:k}, z_{1:k-1}^i) \tag{22}$$

For Algorithm 2,

$$f = \prod_{r=1}^R \left(\prod_{i \neq j} w_{k_r}^{s(i)} \prod_{k=k_{r-1}+1}^{k_r} q_k(z_k^i|x_{1:k}, z_{1:k-1}^i)\right)\left(\frac{1}{N}\prod_{k=k_{r-1}+1}^{k_r} p(z_k^j|x_{1:T}, z_{1:k-1}^j)\right) \tag{23}$$

These densities are normalized, so $\mathbb{E}_g[f/g] = 1$. Thus, our goal is to show $\hat{p}_N(x_{1:T}) = p(x_{1:T})f/g$.

$$p(x_{1:T})\prod_{r=1}^R \frac{(\prod_{i \neq j} w_{k_r}^{s(i)} \prod_{k=k_{r-1}+1}^{k_r} q_k(z_k^i|x_{1:k}, z_{1:k-1}^i))(N^{-1}\prod_{k=k_{r-1}+1}^{k_r} p(z_k^j|x_{1:T}, z_{1:k-1}^j))}{\prod_{i=1}^N w_{k_r}^{s(i)} \prod_{k=k_{r-1}+1}^{k_r} q_k(z_k^i|x_{1:k}, z_{1:k-1}^i)} = \tag{24}$$

$$p(x_{1:T})\prod_{r=1}^R \frac{N^{-1}\prod_{k=k_{r-1}+1}^{k_r} p(z_k^j|x_{1:T}, z_{1:k-1}^j)}{w_{k_r}^j \prod_{k=k_{r-1}+1}^{k_r} q_k(z_k^j|x_{1:k}, z_{1:k-1}^j)} = \tag{25}$$

and pushing in the marginal likelihood

$$\prod_{r=1}^R \frac{N^{-1}\prod_{k=k_{r-1}+1}^{k_r} p(z_k^j, x_k|x_{1:k-1}, z_{1:k-1}^j)}{w_{k_r}^j \prod_{k=k_{r-1}+1}^{k_r} q_k(z_k^j|x_{1:k}, z_{1:k-1}^j)} \tag{26}$$

Now, letting $\alpha_k(z_{1:k}^i) = \frac{p(z_k^i, x_k|x_{1:k-1}, z_{1:k-1}^i)}{q_k(z_k^i|x_{1:k}, z_{1:k-1}^i)}$ one can show that for this sequence of resampling times the weight $w_{k_r}^j$ telescopes into

$$w_{k_r}^j = \frac{\prod_{k=k_{r-1}+1}^{k_r} \alpha_k(z_{1:k}^j)}{\sum_{i=1}^N \prod_{k=k_{r-1}+1}^{k_r} \alpha_k(z_{1:k}^i)} \tag{27}$$

and the estimator $\hat{p}_N(x_{1:T}) = \prod_{t=1}^T \hat{p}_t$ telescopes into

$$\hat{p}_N(x_{1:T}) = \prod_{r=1}^R \left(\frac{1}{N}\sum_{i=1}^N \prod_{k=k_{r-1}+1}^{k_r} \alpha_k(z_{1:k}^i)\right) \tag{28}$$

and thus

$$p(x_{1:T})\frac{f}{g} = \prod_{r=1}^{R} \frac{N^{-1}\prod_{k=k_{r-1}+1}^{k_r} p(z_k^j, x_k | x_{1:k-1}, z_{1:k-1}^j)}{w_{k_r}^j \prod_{k=k_{r-1}+1}^{k_r} q_k(z_k^j | x_{1:k}, z_{1:k-1}^j)} \tag{29}$$

$$= \prod_{r=1}^{R} \frac{N^{-1}}{w_{k_r}^j} \prod_{k=k_{r-1}+1}^{k_r} \alpha_k(z_{1:k}^j) \tag{30}$$

$$= \prod_{r=1}^{R} \frac{1}{N} \left( \sum_{i=1}^{N} \prod_{k=k_{r-1}+1}^{k_r} \alpha_k(z_{1:k}^i) \right) = \hat{p}_N(x_{1:T}). \tag{31}$$

The result follows. An intuitive way to understand this result is the following: Algorithm 2 matches the distribution of every random variable in the particle filter *except* it interleaves a true posterior sample into the set of particles with uniform probability. The only mismatch in the densities are the normalization terms of the resampling probabilities of that privileged posterior sample, with terms $\sum_{i=1}^{N} \prod_{k=k_{r-1}+1}^{k_r} \alpha_k(z_{1:k}^i)$ coming from the filter's resampling and terms $N$ from conditional SMC's resampling. Of course, we never run Algorithm 2, it just serves to define the target density.

**Gradients of $\mathcal{L}_N^{\mathrm{FIVO}}(x_{1:T}, p, q)$.**

We formulate unbiased gradients of $\mathcal{L}_N^{\mathrm{FIVO}}(x_{1:T}, p, q)$ by considering Algorithm 1 as a method for simulating FIVO. We consider the cases when the sampling of $z_t^i$ is and is not reparameterized. We also consider the case where we make adaptive resampling decisions.

First, we assume that the decision to resample is not adaptive (i.e., depends in some way on the random variables already produced until that point in Algorithm 1), and are fixed ahead of time. When the sampling $z_t^i$ is not reparameterized there are three terms to the gradient: (1) the gradients of $\log \hat{p}_N(x_{1:T})$ with respect to the parameters conditional on the latent states, (2) gradients of the densities $q_t$ with respect to their parameters, and (3) gradients of the resampling probabilities with respect to the parameters. All together, the following is a gradient of FIVO,

$$\mathbb{E}\left[ \nabla_{\theta,\phi} \log \hat{p}_N(x_{1:T}) + \sum_{t=1}^{T} \sum_{i=1}^{N} \left( \log \frac{\hat{p}_N(x_{1:T})}{\hat{p}_N(x_{1:t-1})} \nabla_\phi \log q_{t,\phi}(z_t^i | x_{1:t}, z_{1:t-1}^i) + \right. \right.$$
$$\left. \left. \mathbb{I}(\text{resampling at step } t) \log \frac{\hat{p}_N(x_{1:T})}{\hat{p}_N(x_{1:t})} \nabla_{\theta,\phi} \log w_t^i \right) \right] \tag{32}$$

where $\mathbb{I}(A)$ is an indicator function. If $z_t^i$ is reparameterized, then the first and third terms suffice for an unbiased gradient,

$$\mathbb{E}\left[ \nabla_{\theta,\phi} \log \hat{p}_N(x_{1:T}) + \sum_{t=1}^{T} \sum_{i=1}^{N} \mathbb{I}(\text{resampling at step } t) \log \frac{\hat{p}_N(x_{1:T})}{\hat{p}_N(x_{1:t})} \nabla_{\theta,\phi} \log w_t^i \right] \tag{33}$$

In this work we only considered reparameterized $q_t$s, and we dropped the terms of the gradient that arise from resampling.

Second, when the decision to resample is adaptive, the domain of the random variables involved in simulating $\log \hat{p}_N(x_{1:T})$ can be partitioned into $2^T$ regions, over each of which the density is differentiable. Between those regions, the density experiences a jump discontinuity. Thus, there are additional terms to the gradient of $\mathcal{L}_N^{\mathrm{FIVO}}(x_{1:T}, p, q)$ that correspond to the change in the regions of continuity as the parameters change. These terms can be written as surface integrals over the boundaries of the regions. We drop these terms in practice.

**Proof of Proposition 2.**

Assume $p(z_{1:t-1}|x_{1:t}) = p(z_{1:t-1}|x_{1:t-1})$ for all $t \in \{2, \dots, T\}$. We will show $\mathcal{L}_N^{\mathrm{FIVO}}(x_{1:T}, p, q) = \log p(x_{1:T})$ at $q(z_t|z_{1:t-1}, x_{1:t}) = p(z_t|z_{1:t-1}, x_{1:t})$. We will do this by induction, showing that every particle has a constant weight and that $\hat{p}_N(x_{1:T}) = p(x_{1:T})$ is a constant. For $t = 1$ we have

$$\alpha_1^i(z_1) = \frac{p_1(x_1, z_1)}{p(z_1|x_1)} = p_1(x_1) \tag{34}$$

Thus, all particles have the same weight and $\hat{p}_1 = p_1(x_1)$. Now for any $t$ we have that the weights must be $1/N$ since the particles all have the same weight and

$$\alpha_t^i(z_{1:t}) = \frac{p_t(x_t, z_t | z_{1:t-1}, x_{1:t-1})}{p(z_t | z_{1:t-1}, x_{1:t})} \tag{35}$$

$$= \frac{p(z_{1:t}, x_{1:t})}{p(z_{1:t-1}, x_{1:t-1}) p(z_t | z_{1:t-1}, x_{1:t})} \tag{36}$$

$$= \frac{p(x_{1:t})}{p(x_{1:t-1})} \frac{p(z_{1:t} | x_{1:t})}{p(z_{1:t-1} | x_{1:t-1}) p(z_t | z_{1:t-1}, x_{1:t})} \tag{37}$$

$$= \frac{p(x_{1:t})}{p(x_{1:t-1})} \frac{p(z_{1:t} | x_{1:t})}{p(z_{1:t-1} | x_{1:t}) p(z_t | z_{1:t-1}, x_{1:t})} \tag{38}$$

$$= \frac{p(x_{1:t})}{p(x_{1:t-1})} \tag{39}$$

and thus,

$$\hat{p}_N(x_{1:T}) = p_1(x_1) \prod_{t=2}^{T} \frac{p(x_{1:t})}{p(x_{1:t-1})} = p(x_{1:T}) \tag{40}$$

**Implementation details**

We initialized weights using the Xavier initialization [53] and used the Adam optimizer [54] with a batch size of 4. During training, we did not truncate sequences and performed full backpropagation through time for all datasets. For the results presented in Sections 6.1 and 6.2 we performed a grid search over learning rates $\{3 \times 10^{-4}, 1 \times 10^{-4}, 3 \times 10^{-5}, 1 \times 10^{-5}\}$ and picked the run and early stopping step by the validation performance.

**Evaluation and Comparison of Bounds**

Comparing models trained with different log-likelihood lower bounds is challenging because calculating the actual log-likelihood is intractable. Burda *et al.* [12] showed that the IWAE bound is at least as tight as the ELBO and monotonically increases with $N$. This suggests comparing models based on the IWAE bound evaluated with a large $N$. However, we found that IWAE and ELBO bounds tended to diverge for models trained with FIVO.

Although FIVO is not provably a tighter bound than the ELBO or IWAE, our experiments suggest that this tends to be the case in practice. In Figure 3, we plotted all three bounds over training for a representative experiment. All plots use the same model architecture, but the training objective changes in each panel. For the model trained with IWAE, the FIVO and IWAE bounds are tighter than their counterparts on the model trained with ELBO, suggesting that the model trained with IWAE is superior. The ELBO bound evaluated on the model trained with IWAE, however, is lower than its counterpart on the model trained with the ELBO. For the model trained with FIVO, both IWAE and ELBO bounds seem to diverge, but the FIVO bound outperforms the FIVO bounds on both of the other models. As in the figure, we generally found that the same model evaluated with FIVO, IWAE, and ELBO produced values descending in that order.

We suspect that $q$ distributions trained under the FIVO bound are more entropic than those trained under ELBO or IWAE because of the resampling operation. During training under FIVO, $q$ is able to propose state transitions that could poorly explain the observations because the bad states will be resampled away without harming the final bound value. Then, when a FIVO-trained $q$ is evaluated with ELBO or IWAE it proposes poor states that are not resampled away, leading to a poor final bound value. Conversely, $q$s trained with ELBO and IWAE are not able to fully leverage the resampling operation when evaluated with the FIVO bound.

Because of this behavior, we chose to optimistically evaluate models trained with IWAE and ELBO by reporting the maximum across all the bounds. For models trained with FIVO, we reported only the FIVO bound. We felt this evaluation scheme provided the strongest comparison to existing bounds.

Figure 3: Comparison of ELBO, IWAE, and FIVO bounds. We plot the ELBO ($\mathcal{L}$), IWAE ($\mathcal{L}_{128}^{\mathrm{IWAE}}$), and FIVO ($\mathcal{L}_{128}^{\mathrm{FIVO}}$) test log-likelihood lower bounds for a fixed model architecture trained with FIVO (left), IWAE (middle), and ELBO (right). The models are VRNNs trained on the Nottingham dataset with 64 units, $N = 16$, and learning rate $3 \times 10^{-5}$.

**Evaluating TIMIT Log-Likelihoods**

We reported log-likelihood scores for TIMIT relative to an ELBO baseline instead of raw log-likelihoods. Previous papers (e.g., [39, 41]) report the log-likelihood of data that have been mean centered and variance normalized, but it would be more proper to report the results on the un-standardized data. Specifically, if the training set has mean $\mu$ and variance $\sigma^2$ and the model outputs $\hat{\mu}$ and $\hat{\sigma}^2$, then the un-standardized test data would be evaluated under a $\mathcal{N}(\hat{\mu}\sigma + \mu, \hat{\sigma}^2\sigma^2)$ distribution.

Log-likelihoods produced by these approaches differ by a constant offset that depends on $\sigma$. Because the offset is a function of only training set statistics, it does not affect relative comparison between methods. Because of this we chose to report log-likelihoods relative to a baseline instead of absolute numbers. Absolute numbers calculated on standardized data are reported in Tables 3, 4, and 5 to allow for comparisons with other papers.

| N | Bound | Nottingham Train | Nottingham Test | JSB Train | JSB Test | MuseData Train | MuseData Test | Piano-midi.de Train | Piano-midi.de Test |
|---|---|---|---|---|---|---|---|---|---|
| 4 | ELBO | -2.54 | -3.00 | -4.99 | -8.60 | -6.20 | -7.15 | -6.26 | -7.81 |
|  | IWAE | -1.72 | -2.75 | -4.81 | -7.86 | -5.86 | -7.20 | -6.25 | -7.86 |
|  | FIVO | **-1.35** | **-2.68** | **-4.59** | **-6.90** | **-5.64** | **-6.20** | **-5.73** | **-7.76** |
| 8 | ELBO | -2.65 | -3.01 | -4.94 | -8.61 | -5.85 | -7.19 | -6.20 | -7.83 |
|  | IWAE | -1.59 | -2.90 | **-4.47** | -7.40 | -6.23 | -7.15 | -5.71 | -7.84 |
|  | FIVO | **-1.46** | **-2.77** | -5.41 | **-6.79** | **-5.02** | **-6.12** | **-5.69** | **-7.45** |
| 16 | ELBO | -2.06 | -3.02 | -5.08 | -8.63 | -6.22 | -7.18 | -6.71 | -7.85 |
|  | IWAE | -2.12 | -2.85 | -4.86 | -7.41 | -6.54 | -7.13 | -5.17 | -7.79 |
|  | FIVO | **-1.33** | **-2.58** | **-4.45** | **-6.72** | **-5.44** | **-5.89** | **-5.08** | **-7.43** |

Table 3: Train and test set marginal log-likelihood bounds for VRNNs trained on the polyphonic music datasets. We report $\max\{\mathcal{L}, \mathcal{L}_{128}^{\mathrm{IWAE}}, \mathcal{L}_{128}^{\mathrm{FIVO}}\}$ for ELBO and IWAE models and $\mathcal{L}_{128}^{\mathrm{FIVO}}$ for FIVO models. VRNNs trained on the JSB Chorales used 32 units, all other models used 64 units.

| | | TIMIT | | | |
|---|---|---|---|---|---|
| | | 64 units | | 256 units | |
| N | Bound | Train | Test | Train | Test |
| 4 | ELBO | 40,237 | 41,236 | 51,688 | 51,674 |
|  | IWAE | 40,939 | 41,076 | 52,284 | 52,290 |
|  | FIVO | **46,911** | **46,927** | **59,180** | **59,058** |
| 8 | ELBO | 42,892 | 44,007 | 49,872 | 51,055 |
|  | IWAE | 43,713 | 45,213 | 52,827 | 52,859 |
|  | FIVO | **47,343** | **47,259** | **61,080** | **62,685** |
| 16 | ELBO | 43,175 | 42,912 | 51,490 | 51,154 |
|  | IWAE | 43,331 | 44,472 | 53,797 | 54,305 |
|  | FIVO | **48,685** | **49,866** | **61,929** | **62,772** |

Table 4: Train and test set log likelihood bounds for VRNNs trained on the TIMIT dataset with different bounds and numbers of particles. We report $\max\{\mathcal{L}, \mathcal{L}_{128}^{\mathrm{IWAE}}, \mathcal{L}_{128}^{\mathrm{FIVO}}\}$ for ELBO and IWAE models and $\mathcal{L}_{128}^{\mathrm{FIVO}}$ for FIVO models. These results were calculated on data that was standardized (mean-centered and scaled to unit variance) using training set statistics.

| Bound | Nottingham Train | Nottingham Test | JSB Train | JSB Test | MuseData Train | MuseData Test | Piano-midi.de Train | Piano-midi.de Test | TIMIT Train | TIMIT Test |
|---|---|---|---|---|---|---|---|---|---|---|
| ELBO | -2.95 | **-2.40** | -8.68 | **-5.48** | -7.52 | -6.54 | **-7.86** | **-6.68** | **41805** | **40757** |
| ELBO+s | **-2.91** | -2.59 | **-8.64** | -5.53 | **-7.51** | **-6.48** | -7.87 | -6.77 | 40743 | 39832 |
| IWAE | -3.03 | -2.52 | -8.61 | -5.77 | -7.55 | -6.54 | -7.84 | **-6.74** | 42174 | 42226 |
| IWAE+s | **-2.83** | **-2.37** | **-8.15** | **-4.63** | **-7.33** | **-6.47** | **-7.81** | **-6.74** | **44294** | **43387** |
| FIVO | **-2.87** | **-2.29** | -7.06 | -4.08 | **-6.55** | **-5.80** | -7.75 | -6.41 | 49653 | 47748 |
| FIVO+s | -2.92 | -2.34 | **-6.91** | **-3.83** | -6.68 | -5.87 | -7.80 | **-6.34** | **52644** | **50530** |

Table 5: Train and test set log-likelihood bounds comparing smoothing and non-smoothing models. We report $\max\{\mathcal{L}, \mathcal{L}_{128}^{\mathrm{IWAE}}, \mathcal{L}_{128}^{\mathrm{FIVO}}\}$ for ELBO and IWAE models and $\mathcal{L}_{128}^{\mathrm{FIVO}}$ for FIVO models. All models were trained with $N = 4$ and a learning rate of $3 \times 10^{-5}$. The JSB Chorales model used 32 units and the Musedata model used 256 units. All other models used 64 units. TIMIT results were calculated on data that was standardized (mean-centered and scaled to unit variance) using training set statistics.