[Reviews · NeurIPS 2017]

Reviewer 1



This paper generalized the traditional ELBO in Variational Inference to a more flexible lower bound “Monte Carlo Objectives(MCO)”. It modifies IWAE, a previous MCO, to get a new lower bound FIVO based on particle filtering. The paper also gives some theoretical properties of MCO about consistency and bias. Looking beyond classical ELBO is one of most promising directions in study of Variational Inference. However, it also has several problems or unclear explanation: 1. It is better to include more details about the optimization. The influence of biased gradient need more theoretical study. 2. What is the criterion used for resampling; Is it effective sample size? How frequently is the resampling carried out in the experiments? Will frequent resampling increases variance? Why the figure in appendix shows that the variance with resampling is higher than that without resampling? 3. In Section 4.2, it is mentioned that the FIVO assumption cannot guarantee the sharpness. The trade off between this disadvantage and the advantage of FIVO is not thoroughly investigated. 4. How to choose the number of particles for FIVO? Why in figure 2, for IWAE(special case of FIVO without resampling), the largest particle size(N=16) achieves lowest likelihood score?

Reviewer 2



The paper extends Importance Weighted variational objective to sequential model where the posterior is over a sequence of latent variables. In doing so, the authors also present (1) an elegant and concise generalization of IW to the case of any unbiased estimator of the marginal likelihood with a bound on the asymptotic bias, (2) a demonstration that this idea works in the case of non-iid average as in the particle filter with resampling (where particles are coupled through the resampling step). All of these are novel, although it is perhaps not surprising once one realize the implication of (1). Eq (11) seems to deal with the resampling (of discrete particles) through the usual reinforce trick which is known to have high variance. In fact, the authors do observe that particle filter-FIVO gradients computed in (11) has very high variance. However, they should mention this much earlier, right after eq (11), with a heads-up that only the first part of eq (11) is used, and discussion the theoretical implication if possible. For example is the actual gradient estimate no longer unbiased? would it cause any convergence issue? Despite this main limitation, the empirical results seem to show that this approach is worthwhile. However, I would strong encourage the authors to pay more attention to this limitation, be up front about it, discuss potential pitfalls and whether they observed these issues in practice.

Reviewer 3



This paper introduces Monte Carlo objective to improve the scaling of the IWAE and provides a tighter lower bound to the log likelihood than traditional ELBO. Various theoretical properties of Monte Carlo objectives are discussed. An example of Monte Carlo objective is provided to use particle filters for modeling sequential data. The paper is generally well written and I enjoyed reading it. * For the FIVO algorithm, it'd be more self-contained if including a brief description of the resampling criteria or its exact procedure used in the experiment, so that the reference doesn't need to be consulted. * Although not required, it'd be helpful to provide extra examples of Monte Carlo objectives other than applications of sequential data as FIVOs. * I feel it's fairer to stay with a batch of 4 instead of 4N for ELBO to reduce the impact of different batch configurations, even if the computational budget does not match up (line 259). The authors could also consider reporting the running time of each method, and normalize it somehow when interpreting the results. This could also give readers some idea about the overhead of resampling. * I'm a bit worried about the use of the biased gradients to train FIVO. Is it possible that part of the superior performance of FIVO over IWAE somehow benefits from training using the incorrect gradients? It'd be useful to also report the performance of FIVO trained with the unbiased gradients so that the readers will have the idea about the impact of the large variance coming from the resampling term. * It's not clear to me why checking on the KL divergence between the inference model q(z|x) to the prior p(z) addresses the need to make use of the uncertainty (line 279 to 283). What's the connection between "the degree to which each model uses the latent states" and the KL divergence? How to interpret ELBO's KL collapses during training? Doesn't the inference model collapses onto the prior suggest that it carries less information about the data? * Typo: it should be z ~ q(z|x) in line 111-112.